# Changes in Fabric Surface Pilling under Laser Ablation

**DOI:** 10.3390/s20205832

**Published:** 2020-10-15

**Authors:** Ewa Korzeniewska, Jarosław Gocławski, Joanna Sekulska-Nalewajko, Maria Walczak, Bożena Wilbik-Hałgas

**Affiliations:** 1Institute of Electrical Engineering Systems, Lodz University of Technology, 90-924 Lodz, Poland; maria.walczak@p.lodz.pl; 2Institute of Applied Computer Science, Lodz University of Technology, 90-924 Lodz, Poland; jaroslaw.goclawski@p.lodz.pl (J.G.); joanna.sekulska-nalewajko@p.lodz.pl (J.S.-N.); 3Institute of Security Technologies “MORATEX”, 90-505 Lodz, Poland; bhalgas@moratex.eu

**Keywords:** laser, textile modification, pilling, fractal dimension, optical coherent tomography, computer image analysis

## Abstract

Textiles require finishing to improve their usability and functionality but in the first place, to reduce of pilling tendency, which affects all kinds of synthetic and natural fabrics. Several laser ablation tests have been applied to the selected fabrics with different chemical composition to reveal the impact of this process on the pilling behavior. To reflect the pilling changes, two textural descriptors have been proposed to textile images obtained with optical coherence tomography (OCT). They showed the trend to reduce values with increasing laser power applied to the tested fabrics. It has been demonstrated, that in the case of textiles based on polyester threads, laser modification of the product surface led to a significant reduction in their tendency for pilling.

## 1. Introduction

Laser technologies are widely used in the textile industry, having replaced such conventional dry surface modification techniques such as sandblasting and intentional aging, which may have even been destructive. Conventional fabric modification processes are potentially hazardous and may be harmful to the environment. The laser technique makes it possible to modify the surfaces of fabrics without the use of water and often without toxic chemicals. Lasers can be used to mark or engrave textile materials, to identify or add information about the product, to create a characteristic logo, and even to prevent theft [1]. The possibility of combining several processes, such as surface modification, cutting, engraving, and marking in one production cycle, is an undoubted advantage of using laser technologies in the clothing industry [2,3]. They can be even used to join the wearable electronics with textiles [4].

Laser modification is a competitive method in comparison to conventional denim cutting, embossing, engraving, and fading processes. The probability of damaging the product is low and there is no need for additional consumables. Unlike chemical technologies, when a laser is used to treat the surfaces of fabrics there are no toxic by-products [2]. Unlike mechanical tools, a laser beam does not show signs of wear after multiple uses, so it can provide greater precision. Physical melting of the treated surface prevents fraying of the treated fabric. There is no risk of contamination and lasers can be used to treat almost any textile [3]. Due to the stability and easy to use nature of the laser source, it is possible to precisely treat the surface while maintaining a zone of minimal heat influence in the fabric structure [5]. Laser as well as plasma ablation can be also used to change the wettability of the textile [6].

One of the most important ways of describing textiles is fabric hand. This term is used to describe the perceived overall aesthetic quality of a fabric. The quality of a textile product is mainly determined by touch [7]. The feel of a fabric has a direct impact on the attractiveness of the product. According to Dudeja [3], laser treatment of the surface of fabrics can be applied without compromising the hand of the fabric. It does not change the stiffness, softness, smoothness, or ease of draping of the fabric, and does not affect the aesthetic appearance of garments. Indeed, cotton fabrics after laser treatment can have better drapability and wrinkle recovery. Dudeja also found that fabrics with a mixed cotton/polyester composition have higher stiffness after laser treatment, which may be useful in the production stage of some elastic fabrics.

The formation of pilling results mainly from the properties of the raw material used in the production of textile products, the number of twists in the yarn, as well as the gauge number. Fabrics or knitwear made from long and hard yarn are more susceptible to pilling. The thickness of knitted fabrics composed of low-twist yarns is also an important factor. The fluffier the yarn, the more prone the fabric is to pilling. In a product with a loose fiber configuration, the tendency to form pills is greater [8]. The denser the structure of the product, the lower the susceptibility to pilling.

The phenomenon of pilling is important not only for aesthetic reasons, but also due to changes in the surface morphology, which is very important in the case of modern technologies related to wearable electronics. The production of new types of materials and the modification of textiles is currently an explored area, especially in the context of using them as a substrate for elements of wearable electronics or textronics [9], or the production of wearable sensors [10,11]. The process of pilling can be divided into three stages [8]. In the first stage, fibers come to the surface of the material due to mechanical factors. In the second stage, the threads become entangled, and either attach to the surface of the product or detach in the third and final stage. As the fibers exit the threads, they do not increase in length. However, new fibers are extracted from the threads, which connect with the already protruding threads or piles on the surface of the product. According to Gintis and Mead [8], the largest number of pills arises within the first five minutes of the forced pilling process. After this time, the increase in the mass of extracted fibers and the appearance of new spheres are much lower. According to Gintis and Mead, viscose fiber products tend to self-clean, due to the low bending resistance of the fiber.

More recently, researchers have extended the three-stage model of pilling with further stages. According to Cooke and Ukponmwan [12,13], the process of pilling formation includes:extraction of loose threads from the yarn that makes up the fabricthe generation of pills, also called combing the fiberspill growthshaping the structure of the pillschanges inside the core and ligament of pillsdetachment of pills.

Gintis and Mead [8] conclude that the formation of pilling primarily involves the longest fibers. These become, at a later stage, the ligament of the core. The formation of the core also depends on the presence of external factors, such rubbing against other elements.

According to [12], when fibers are combed, they wrap around each other, due to their rubbing against each other (Figure 1). In the combing process, the fibers are subjected to repeated bending and stretching, which causes weakening and finally breakage. The detached fibers, interwoven with each other, form the nuclei of pills. The closer they are to the center of the nucleus, the more the fibers stick, whereas the fibers on the surface of the sphere break faster. Cooke et al. [9] observed the formation of hooks at the ends of some protruding fibers, which may initiate the process of entanglement. When fibers become entangled, the threads are mechanically damaged. For fibers with a diameter greater than 30 µm, this damage results mainly from repeated bending. Fibers with smaller diameters tend to migrate from the yarn structure, and in the case of fibers with diameters of less than 20 µm this is the dominant phenomenon [12]. Pilling tendencies can be reduced by the physicochemical process of finishing textile products; for example, by using post-dyeing and chemical finishing, such as chlorination [14], argon low temperature plasma [15], peroxycarboximidic acid oxidation, and selective enzyme digestion of wool scales [16]. They may also be finished with polyurethane resin [17], among other methods [7].

In this paper, we propose the use of a laser beam to modify the fibers on the surface of a textile product, thereby reducing its tendency to pilling. In our opinion, the commonly used method of visual assessment is too subjective [18]. Therefore, we use a newly developed method for computer assessment of pilling, using optical coherent tomography (OCT), which we presented in a previous publication [19]. Optical coherence tomography is a noninvasive imaging technology based on low-coherence interferometry [20,21,22]. It can provide cross-sectional images of different materials that are transparent or translucent to the infrared light of the scanning source. The system uses a Michelson interferometer, which compares the interference signals of light backscattered from a sample to the light reflected from a reference mirror. Unlike other kinds of tomography, such as electrical impedance, the OCT system can be based in the time-domain (TD-OCT) or Fourier-domain (FD-OCT) [23,24]. In TD-OCT, a depth scan is performed by moving the reference mirror. In FD-OCT, the scan is carried out using a narrowband laser swept source (SS-OCT) [25] or broadband spectral domain laser source (SD-OCT) [26] with a diffraction grid and spectrometer. This imaging technology is used for many biomedical applications, including ophthalmology [27], stomatology [28], cardiology [29], and dermatology [30]. Optical coherence tomography can also be applied as a tool for testing surface roughness [31], composite material layers [32], food products [33], printed circuit boards [34], and solar cells [35], as well as in polymer science [36]. In textile sciences, OCT has been used to sort fabrics [37], assess industrial laundry cleaning [38], and evaluate the degree of fabric pilling [19]. Currently, advanced TD-OCT and FD-OCT optical systems are quite expensive, which limits their usage.

## 2. Materials and Methods

### 2.1. Textile Characteristic

Research was conducted on three different types of knitted fabrics commonly available on the market and widely used in the clothing industry. The compositions and characteristics of the fabrics are presented in Table 1. All the tested knitted fabrics had a surface density of 240 g/m^2^ and were considered to have a high degree of resistance to unwanted pilling. They differed in terms of the chemical composition of the fibers. Knit fabric F1 was made entirely of polyamide, F2 was a mixture of polyester and polyacrylonitrile in ratio of 65:35, respectively, and F3 was cotton with an almost 32% polyamide admixture.

The knitted fabrics were subjected to laser treatment to modify the surface and model their pilling tendency.

### 2.2. Process of Laser Modification

The experimental set-up consisted of a fiber-pulsed laser with a wavelength of 1060 nm and pulse duration set to between 15 ns and 220 ns. The nanosecond duration of the pulses enabled the thermal treatment to be very precisely localized on the top layer of the textile. The laser beam radiation was controlled by a computer, using five variables: pulse duration, overlapping, hatching, radiation power, scanning velocity of the beam, and frequency of pulse repetition. The mode of operation of the laser is illustrated in Figure 2. Based on the symbols in Figure 2, the overlapping *O* in laser radiation can be defined as
(1)$$O=\frac{D-x}{x},\;\;\;x=\frac{v}{f}\;\;$$
where *D* is the laser beam diameter, *x* denotes the beam offset, *v* is the laser scanning speed, and *f* is the pulse repetition frequency. If the speed is diminished together with the frequency of repetition, the distance between the overlapping spots of the laser can be maintained at the same level (Figure 2). Depending on the purpose of the technological process (e.g., welding, surface modification, or cutting) or the required structure [39,40], the overlap value is different. For example, in the case of precision cutting, the overlap is from 0.75 to 0.9 [41]. Given the structure of the knitted fabric and the mutual arrangement of threads in the textile product, for surface modification the overlap value was set to *O* = 0.01. Hatching was set to 0.01 mm to ensure that the largest possible surface area of the fabric was modified.

It is impossible to guarantee that the laser will always have the same effect on the textile, because the time of the interaction at the spot may be prolongated, which can cause an increase in heat at the affective zone. A stronger reaction between the laser beam and the textile can also be observed when the radiation power and pulse duration are increased. These very complicated phenomena should be taken into account when selecting the parameters for laser beam radiation, which can cause visible thermal damage to the sample, beginning with surface bronzing and melting (Figure 3c,d). All treatments were performed in normal air atmosphere. Using argon does not improve the results of antipilling.

Thermal influence can increase at slower scanning speeds. Such radiation may not only remove the thinner thread above the surface of the textile, but also damage deeper layers. To choose the proper parameters of the laser beam, several tests were performed. The range of the parameters of the laser were chosen based on previous studies [19]. The focusing lens and F-Theta objective allows a single mode of laser beam radiation to be obtained, in the shape of a gaussian mode. We used SPI Nd-YAG laser (Southampton Photonics Inc., Southampton, UK) with power up to 18 W, pulse duration 220 ns and speed of scanning up to 2000 mm/s. Table 2 presents the range of values for the parameters.

How the laser beam radiation is scanned onto the surface of the textile plays important role in decreasing the tendency of pilling. For the best results, the diameter of the beam should be in good correlation with the hatching. Whether the laser beam is guided in one direction or in two perpendicular directions is also very important. In the present study, the textile surface was modified with a laser beam in the direction of the rows and columns of the knitted textile product, either vertically up and down or horizontally left and right [42]. The values for power and the corresponding energy of the laser beam are presented in Table 3.

### 2.3. Methods of Pilling Assessment

Three different pilling simulation methods were used for the three different types of polyester fabrics:Manual test T1, in which the knitted fabric was rubbed with the speed 23–25 cm/s for 15 s with a harsh hard fiber brush at a constant pressure of 2 N controlled by a strain gauge;Manual test T2, in which the knitted fabric was rubbed with the speed 23–25 cm/s for 15 s with an unglazed ceramic plate at a constant pressure of 1 N;Martindale’s test MT, carried out in accordance with the Polish Norm introducing the International Norm PN-EN ISO 12945-2: 2002 standard [18] for 5000 friction cycles, at a constant pressure of 100 N, diameter 20 cm.

T1 and T2 cause the spinning thread of the fiber component to protrude above the fabric surface. This is the initial stage of the pilling process. The MT test, which corresponds to a certain period of use, caused the protruding fibers to fold into pills. This made it impossible to observe the behavior of the fibers during the initial stage of use. The samples after the MT test were assessed by three independent experts using the commonly accepted five-point scale presented in Figure 4 [43].

### 2.4. Textile Image Acquisition

A Spark OCT-1300 image acquisition system (Wasatch Photonics Inc., Morrisville, NC, USA) was used to register the infrared spatial images of the measured fabric layer. A block diagram of the system is shown in Figure 5. The device consists of three main modules: an OCT Engine, an OCT Imaging Probe, and a Computational Engine. The OCT Engine contains a broadband laser source of low coherence infrared light with a wavelength of around 1300 nm, a beam-splitter, an interferometer reference arm, and a spectrometer transmitting an image of the material heterogeneity from the diffraction grating to the imaging module of the Computational Engine. The scanning depth can be calculated directly from the acquired Fourier transform spectra in the diffraction grating, without moving the reference arm. The OCT Imaging Probe contains a scanning mirror, optics, and a color camera for creating en-face visible-light images of the scanned region, illuminated by an additional visible light source.

The computational engine is a PC, including a digital frame-grabber for acquiring spectrometer data from the OCT engine module. Once the infrared laser light has penetrated the region of the textile fabric, the intensity of the signal reflected in the location (*x,y,z*) is stored in the PC memory in the form of a three-dimensional image array.

Each OCT volumetric image is acquired in a raster of 512 × 512 × 640 voxels in the Cartesian coordinate system, where each voxel has dimensions equal to *dx* = 10.2 μm, *dy* = 9.6 μm, and *dz* = 5.4 μm. Therefore, the image covers a volume of 0.5 × 0.5 × 0.4 cm, containing both the fabric layer and the pilling layer above. The scanning process collects a spatial image from the B-scan frames (in OXZ-planes), acquired in real time. This takes around 20 s, including the time needed to save the resulting image file. To facilitate detection of the fabric layer inside the OCT image, it is assumed that the layer is maintained in a horizontal orientation inside each B-scan.

### 2.5. Assessment of Pilling Texture

To assess pilling intensity, the fabric image was subjected to textural analysis, providing quantitative indicators of pilling. The custom measurement algorithm shown in Figure 6 was developed for the Python 3.6 environment running on the Windows 10 operating system [44]. The firmware enables OCT image acquisition of fabrics and registers the results in DICOM image files, which are then loaded as three-dimensional arrays into the measurement program workspace. The first stage of the OCT image processing algorithm is shown Figure 6, block 2. This stage is responsible for extraction of the pilling layer *L_P_* located above the fabric surface and regarded as a separated image *J*(*x,y,z*). We have described the detection principle of this layer in a previous article [19].

The next step of the algorithm, block 3, is median filtering [45] of the acquired OCT image *I*, expressed in Equation (2):(2)$$I\left(p\right)=median\left\{I\left(q\right)\right\},\;q\in W_{M}\left(p\right),\;p,q\in I$$
where $W_{M}\left(p\right)$ denotes the cuboid window of dimensions [$W_{X}\times W_{Y}\times W_{Z}$] around each voxel p=(x,y,z) of the image *I*. This is a necessary step before pilling analysis, to reduce the speckle noise inherent in the content of OCT images. Filtering is performed by a Python script applying the local window of [5×5×5] voxels.
(3)$$J\left(p\right)=I\left(q\right),\;q\in L_{P},$$
where the layer image *J*(*p*), *p* = (*x,y,z*) is formed as part of the denoised image *I* using pixels *q*, which belong to the pilling layer *L_P_* located above the fabric surface.

In the presence of pilling, OCT infrared radiation is reflected from fibers protruding from the fabric surface, manifested in the pilling layer image *J* as many regions of pixels brighter than their surroundings. The number of these regions and the space they occupy increase with the amount of migrating material, in the form of individual fibers or bundles of fibers. Therefore, a possible measure of the degree of pilling may be the number of bright pixels in the image *J*. The OCT image *J*, previously denoised by the local median filtering in Equation (2), is binarized with a global brightness threshold according to Equation (4) (Figure 6, block 4):(4)$$J_{B}\left(x,y,z\right)=\{\begin{array}{ll} 1, & \;J\left(x,y,z\right)>T_{OTSU}\\ 0, & \;\;otherwise \end{array}.$$

The computed Otsu threshold [46] *T_OTSU_*(*I*) is evaluated based on the intensities of the denoised image *I*(*x,y,z*). The thresholding in Equation (4) determines the fraction *f_P_* of pixels with brightness similar to the fabric layer. This corresponds to the outlying fibers of pilling in the *L_P_* layer. The fraction *f_P_* is evaluated as
(5)$$f_{P}=\frac{1}{V_{P}}\underset{x,y,z_{P}}{\sum }J_{B}\left(x,y,z_{P}\right),$$
where *z_P_* ∈ *L_P_* belongs to the layer *L_P_* and *V_P_* is the volume of the layer expressed in voxels.

We also propose the fractal dimension *D* [47,48] of the pilling layer image *J_B_* as another fabric pilling measure. This feature of the image texture increases with *J_B_* image space filled by protruding fibers, which also increases *J_B_* internal complexity. The fractal dimension is evaluated for the boundary image *J_E_* obtained as the exclusive disjunction of *J_B_* and its morphological erosion with the spherical structuring element $S_{E}$ of the unit radius (Equation (6)) [49,50].
(6)$$J_{E}=\;J_{B}\;\underline{\vee }\left(J_{B}⊖S_{E}\right),\;\;\;$$
where $\underline{\vee }$ denotes the exclusive disjunction symbol and ⊖ is a morphological erosion operation. The proposed fractal dimension *D* is the box-counting (Minkowski) dimension $dim_{box}\left(S\right)$ applied in space according to the formula
(7)$$dim_{box}\left(S\right)=lim_{\epsilon \to 0}\frac{\log N\left(\epsilon \right)}{\log\left(1/\epsilon \right)},\;\;\;$$
where ϵ denotes edge length of the sliding cube, S is the 3D data set of image voxels. The dimension is estimated by least squares method through a linear equation: Y=D×X+A where Y = log(N) and X =log (1/ϵ) [51]. To obtain the dimension *D*, a gliding box of size $\left[\epsilon_{i}\times \epsilon_{i}\times \epsilon_{i}\right]$ is used. The number of boxes $N(\epsilon_{i})$ containing at least one bright pixel in the image *J_E_* is computed for each box of size $\epsilon_{i}=s,s/2,s/4,\ldots ,2$ starting with the initial size *s* equal to the smallest image dimension rounded to the power of two.
(8)$$y_{i}=\log\left(N\left(\epsilon_{i}\right)\right),\;\;\;x_{i}=\log\left(\epsilon_{i}\right),$$

The pairs $\left(x_{i},\;y_{i}\right)$ in the box data are computed according to Equation (8) and then the *polyfit* least squares method for polynomial approximation [52,53] is used, in Equation (9), to fit a line to the points:(9)$$a,b=polyfit\left(\left[x_{i}\right],\left[y_{i}\right],deg=1\right),\;\;$$
where $\left[y_{i}\right],\left[x_{i}\right]$ are arrays of the point coordinates, *deg* = 1 is the polynomial degree and *a* and *b* are the coefficients of the regression line approximated on a double logarithmic scale for $N\left(\epsilon_{i}\right)$ and 1/$\epsilon_{i}$. The fractal dimension is computed in Equation (10) as the negative slope of the line.
(10)$$f_{D}=-a$$

### 2.6. Statistical Analysis

A one-way ANOVA was used for analysis of variance in the pilling indicators *f_p_* and *f_D_* for the sample groups. It was also used to determine whether any of the group means were significantly different statistically depending on the laser power used. ANOVA analysis was preceded by verification of the data series for the assumptions of normality in the value distribution and the homogeneity of variance, using the Shapiro–Wilk test [54] and the Brown–Forsythe test [55], consecutively, at a level of significance α = 0.05. The omega-squared (*ω*²) coefficient was used as an effect size measure to describe the part of the variance in the pilling indicator that is explained by the predictor (laser power used for fabric treatment) [56].

After the analysis of variance, the trend in the differences between subsequent pairs of means for a given pilling index in a series of measurements after each abrasion test was investigated using the contrast least significant difference (LSD) Fisher test [57] for linear trend. Because the results were inconclusive, a two-sided *p*-value was used and a significance level α = 0.05. All statistical analyses were performed using PQStat software (PQStat Software Inc., Poznań, Poland). In addition to the Fisher LSD test, using the Excel trend line tool trend lines were added to the point charts presenting the average values of the pilling indicators.

## 3. Results

Laser ablation was applied to the surface layer of three fabrics with various chemical compositions to improve their anti-pilling behavior. Three different abrasive tests were performed to determine the effects of laser treatment. The manual tests T1 and T2 differed in terms of friction force, and the third test was the Martindale test MT. After the MT test, pilling grades were determined on a five-point scale. The results are presented in Figure 7. The pilling level of knitwear F1 produced from polyester was assessed as five, regardless of the laser beam power used to modify the surface structure. During the test, this textile was observed to be self-cleaning. Both knitwear F2 (polyester-polyacrylonitrile) and knitwear F3 (cotton-polyamide) were scored two on the pilling scale before laser treatment. After low-power laser treatment (8W), the experts noticed the uneven appearance of saws on the surface of the samples, especially in the central area. They rated the susceptibility to fuzz formation on the surface of the samples as three on the pilling scale. The rating did not change regardless of the level of laser power used.

After each test, the fabric surface was scanned using an infrared laser beam to obtain volumetric OCT images, which were then analyzed to assess pilling intensity using the texture attributes of the pilling layer visible above the fabric surface. Sample OCT images of the tested fabrics before and after abrasion tests are shown in Figure 8. Pilling symptoms were assessed by evaluating the white pixel fraction *f_P_* defined in Equation (5) and the fractal dimension *f_D_* described in Equation (9).

The texture characteristics of the pilling layer after laser ablation are shown in Figure 9. It can be seen, that the pilling behavior of the knitwear varied depending on the fabric type and the abrasion test used. In general, modifying the fabric surface reduced the pilling level noticeably in the case of fabrics produced from polyester and polyester/polyacrylonitrile yarns. However, this reduction only became apparent after manual abrasion tests, which cause the initial phase of pilling.

According to the variance homogeneity test, which was performed to determine whether the variance of the groups was equal or not, the differences between the groups after the manual abrasion test were usually statistically significant. As shown in Table 4 and Table 5, the main effect of laser ablation was significant for the polyester fabric after the T1 test, in terms of both *f_P_* and *f_D_* descriptors, where F(5,30) = 17.75, *p* < 0.000001, *ω*^2^ = 0.70 and F(5,29) = 27.07, *p* < 0.000001, *ω*^2^ = 0.78. Among the fabrics tested, the observed effect size index *ω**^2^* reaches the highest value in the case of the polyester fabric, indicating a strong relationship between laser ablation and the occurrence of pilling symptoms.

In the case of fabric F2, a hand test with higher friction force revealed a strong laser effect on the pilling texture features, where F(6, 48) = 8.55, *p* < 0.00001, *ω*^2^ = 0.45 and F(6, 48) = 8.47, *p* < 0.0001, and *ω*^2^ = 0.41 for the *f_P_* and *f_D_* descriptors, respectively. The values of the *ω*^2^ effect coefficient indicate that fabric modification with the laser explains 45% of the pixel fraction variability and 41% of the pilling layer fractal dimension variability observed in the OCT images.

There was a clear downward trend in the appearance of pilling, as shown in the *F*-statistics from the LSD Fisher test of the assumed linear trend (Table 4 and Table 5). With the highest laser power used, decreases in the pixel fraction of around 74% and 49% were noted for the polyester fabric and the polyester/polyacrylonitrile fabric, respectively. Simultaneously, the fractal dimension of the pilling layer of these fabrics decreased, by 17.6% and 8.4% respectively. The third type of fabric, containing cotton and polyamide, showed very variable pilling behavior under increasing laser ablation power, and in general a weak reduction in pilling, which may be associated with the non-susceptibility of natural fibers to the relocation and deposition of fiber nanoparticles. In the manual tests, a reduction of the pixel fraction of outlying fibers was observed, of about 20% and 25% in the tests with high and low friction force, respectively. However, only the trend obtained with the weaker manual test was statistically significant, F(8, 18) = 6.03, *p* < 0.0122, *ω*^2^ = 0.60. The decrease in the fractal dimension was very small (2%), and no statistically significant downward trend was observed.

The Martindale test, which leads to the advanced pilling stage, did not reveal a reduction in pilling for any of the tested fabrics. The values of the tested pilling descriptors exhibited either a horizontal or slightly rising trend, confirming the visual inspection of the knitwear in the OCT images. In the case of polyester fabric F1, the standardized test led to the mass detachment of molded pills and loose fibers. Fiber breakage occurred in the case of both the original textile and the textile modified by laser ablation. However, this phenomenon decreased with increasing laser power, as shown by the trend lines (Figure 9). There were statistically significant differences in variance for both descriptors, *f_P_* F(5, 14) = 3.93, *p* < 0.0473, ω^2^ = 0.30 and *f_D_* F(5, 14) = 6.93, *p* < 0.0052, and ω^2^ = 0.52. The effect size *ω*^2^ measure indicates that surface modification by the laser explains 30% of pixel fraction variability and 52% of fractal dimension changeability. This may suggest that the fibers become more resistant to breakage with increasing laser power, and thus the self-cleaning by this polyester fabric is less effective during intensive testing.

## 4. Conclusions

This article has presented a new method for the reduction of fabric pilling, using laser ablation. Three types of fabric were tested in this research. The action of the laser beam on the fabric caused local melting on the surface of the fibers of the threads from which the textiles are made. Even at low laser power, the surface of the knitted or woven fabric was modified. This contributed to reduce the tendency for individual threads to become loose and protrude from within the product. The tendency for the threads to become entangled and form undesirable fuzz, commonly noticed as pilling, was thereby reduced. Excessive laser power caused inordinate melting of the protruding fibers, and due to the detrimental aesthetic effects rendered the textile product unsuitable for use.

From OCT images of the fabric surfaces, we observed a reduction in pilling as the laser energy increased. The reduction in pilling was further investigated using two manual methods and the Martindale method. The method of computer-assisted OCT image analysis developed and presented previously by the authors was adapted to quantify changes in fabric pilling as a result of laser ablation. This method uses two quantitative pilling descriptors: the fiber fraction and the fractal dimension of the pilling layer determined above the fabric surface. After manual abrasion tests, both descriptors followed a downward trend with increasing laser ablation power. Changes in their values express a reduction in fabric pilling. The Martindale test showed no reduction in pilling.

The descriptor values computed for the polyester fabric remained low, independent of laser power due to the self-cleaning effect. The two other tested fabrics, produced from polyester/polyacrylonitrile and cotton/polyamide, maintained constant levels for the descriptors on the laser power scale, despite the fact that the classic rating on a five-point scale showed a pill reduction of 1 degree. The reason for this discrepancy may be the low precision of fuzz estimation by the experts, which in OCT images can be measured more accurately. This suggests that the proposed approach may be complementary to the classic method of pilling evaluation. The pilling descriptors can be applied only when the density of fibers in the pilling layer is much lower than that in the fabric structure.

The laser method for the reduction of pilling presented in this paper can be used to treat almost any textile, and has several advantages over popular chemical methods, including the fact it leaves no toxic residues, allows easy control of the heat source, and requires relatively low energy for the removal of pills. The disadvantage of this approach may be the increased stiffness of some fabrics after laser treatment. The maximum ablation energy must also be determined separately for each type of fabric.

## Figures and Tables

**Figure 1 sensors-20-05832-f001:**
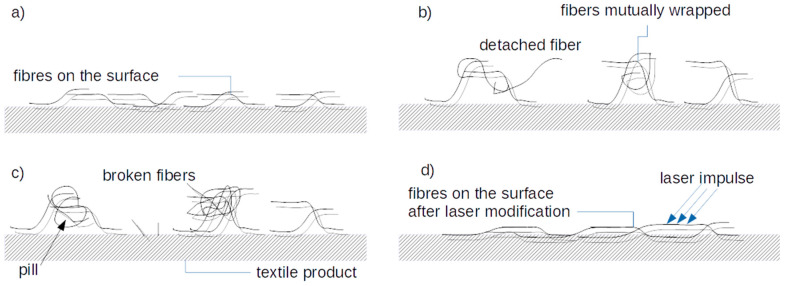
Mechanism of pilling formation, (**a**) threads emerge from inside the fabric; (**b**) extraction of loose threads; (**c**) pills are generated; (**d**) fabric behavior after laser treatment (melted, connected threads).

**Figure 2 sensors-20-05832-f002:**
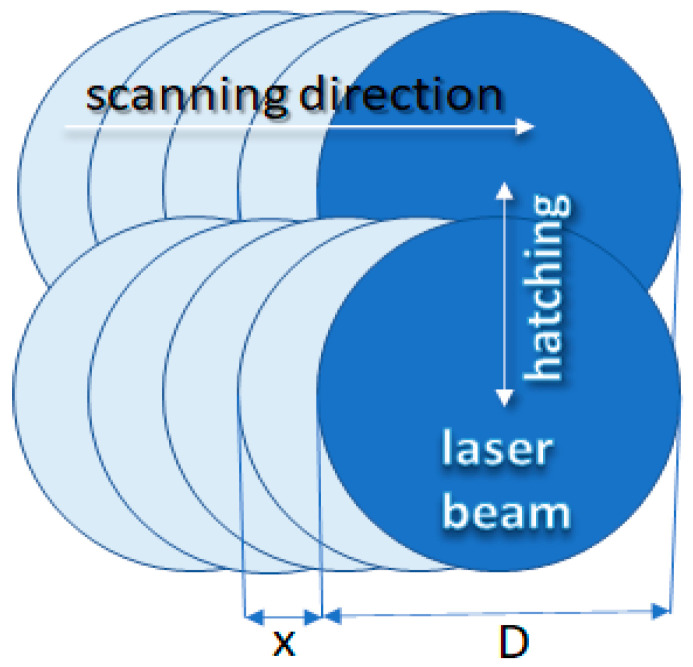
Mode of operation of the laser during textile modification: D=26 μm—laser beam diameter, x≈0.99D—beam offset assuming *O* = 0.01 (according to Equation (1)).

**Figure 3 sensors-20-05832-f003:**
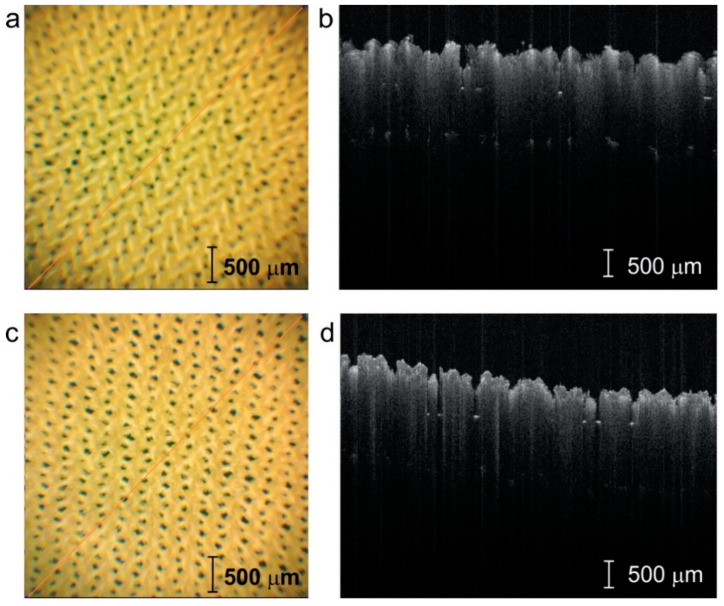
Microscopic photos and optical coherence tomography (OCT) cross-sections of the laser-modified surface of F1: (**a**,**c**)—camera view, (**b**,**d**)—OCT view; upper row (**a**,**b**)—0 W, lower row (**c**,**d**)—16 W.

**Figure 4 sensors-20-05832-f004:**
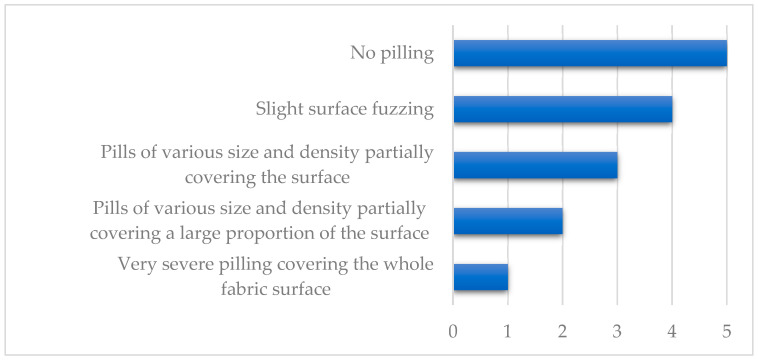
Pilling scale as described in standard PN-EN ISO 12945-2:2002 [18].

**Figure 5 sensors-20-05832-f005:**
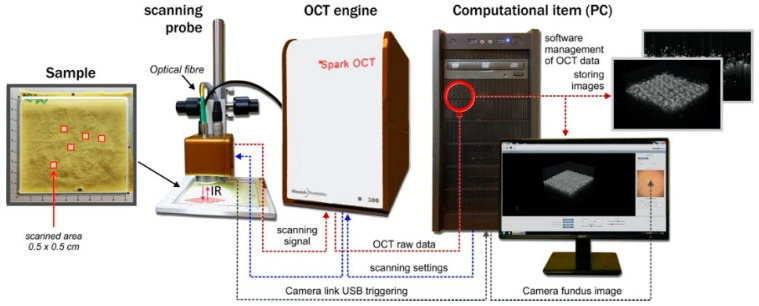
Diagram of the Spark OCT 1300 nm scanning system.

**Figure 6 sensors-20-05832-f006:**
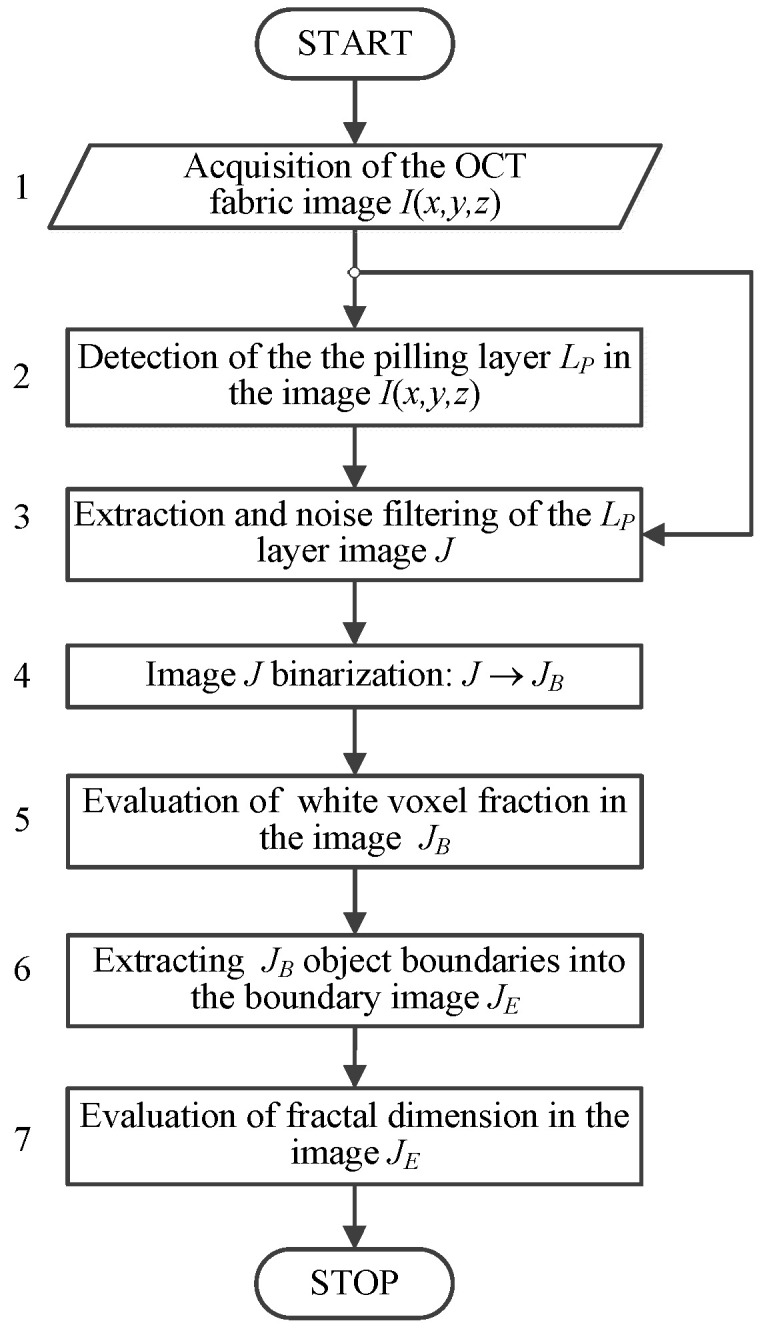
Flowchart of the applied algorithm: *I*—acquired fabric OCT image; *J*—image of the pilling layer *L_P_*; *J_B_*—result of image *J* binarization, *J_E_*—image of *J_B_* boundary objects.

**Figure 7 sensors-20-05832-f007:**
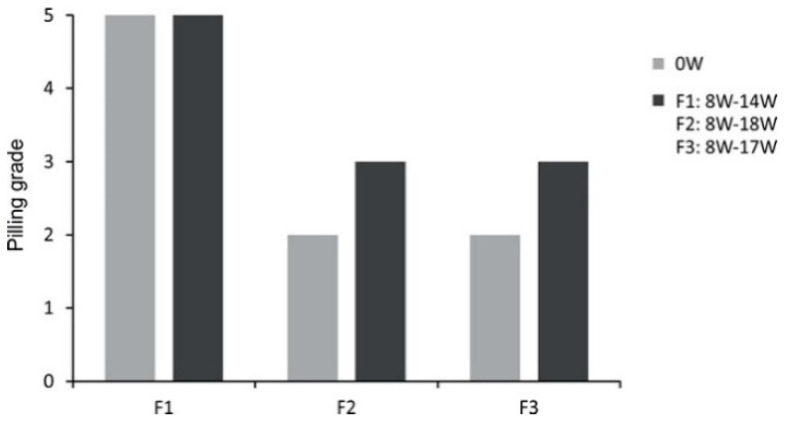
Pilling grade of textiles determined after the Martindale test MT, based on the five-point rating scale (PN-EN ISO 12945-2:2002) [18].

**Figure 8 sensors-20-05832-f008:**
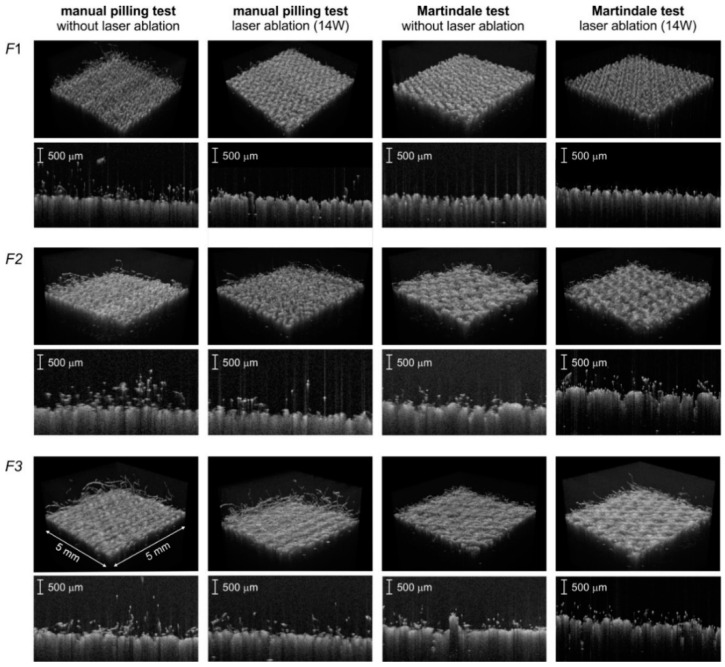
Perspective view of the fabric surface and two-dimensional fabric cross-section obtained by OCT, illustrating the appearance of pilling phenomena after abrasion tests T1 and MT. Fabric samples not subjected to laser treatment and subjected to laser ablation with 14 W laser power. The scale of volumetric images is common to all sub images.

**Figure 9 sensors-20-05832-f009:**
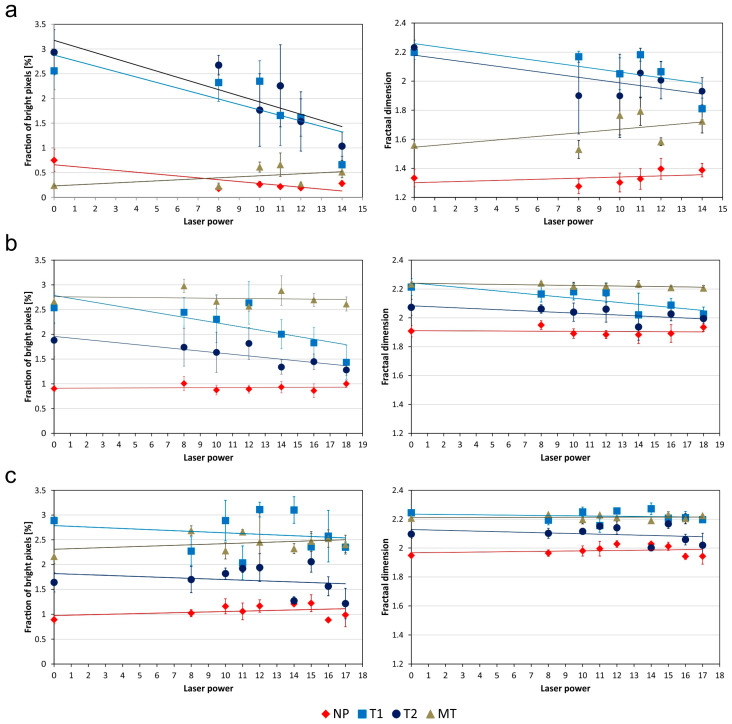
Plots of the texture descriptors of the pilling layer of the tested fabrics subjected to laser ablation, including both mean feature values and approximated trend lines for the fiber pixel fraction (**left column**) and fractal dimension (**right column**), as the laser power increased. Samples: (**a**), (**b**) and (**c**)—fabrics F1, F2, and F3; not pilled samples (*NP*), after manual abrasion test T1 and test T2, after Martindale test (MT).

**Table 1 sensors-20-05832-t001:** Parameters of the knitwear samples.

Name	Composition	Type	Surface Mass
			(g/m^2^)
F1	100% polyester	single jersey, left–right weave	240
F2	65% polyester, 35% polyacrylonitrile	Lacoste blue	240
F3	68% cotton, 32% polyamide	smooth knit, left–right weave	240

**Table 2 sensors-20-05832-t002:** Parameters of the radiation beam of the fiber laser.

	Wavelength	Pulse Duration	Pulse Energy	Pulse Frequency	Hatching Distance	Scan Speed	Beam Diam.
	(nm)	(ns)	(µJ)	(µHz)	(µm)	(mm/s)	(µm)
range	1060	15–220	88–198	35–290	10–40	200–2000	26
test	1060	220	88–198	35	10	400	26

**Table 3 sensors-20-05832-t003:** Laser beam energy and power used to modify the textile surface.

**Energy**	(µJ)	88	110	121	132	154	165	176	188	198
**Power**	(W)	8	10	11	12	14	15	16	17	18

**Table 4 sensors-20-05832-t004:** Analysis of variance (ANOVA) of the pixel fraction for textiles after laser modification and the Fisher least significant difference (LSD) test for linear trend. NP—not pilled samples, T1, T2—manual abrasion tests, MT—Martindale test. * denotes the Brown–Forsythe correction of F-statistic, n.s. not significant data.

Textile	Test	ANOVA Results			Fisher LSD Test	
		F	P	*ω*²	F	p
F1	T1	17.749 *	<0.000001	0.699	60.940	<0.000001
	T2	4.670 *	0.0094	0.372	9.887	0.0020
	MT	3.932 *	0.0473	0.304	2.579	0.0653
F2	T1	8.555	<0.00001	0.452	46.904	<0.000001
	T2	3.690	0.0040	0.218	10.280	0.0012
	MT	2.357	0.0871	0.279	0.973	0.1703
F3	T1	5.004	0.0022	0.543	0.843	0.1853
	T2	6.005	0.0008	0.597	6.034	0.0122
	MT	1.758	0.2988	0.194	n.s.	n.s.

**Table 5 sensors-20-05832-t005:** Analysis of variance (ANOVA) of the fractal dimension for textiles after laser modification and the Fisher LSD test for linear trend. NP—not pilled samples, T1, T2—manual abrasion tests, MT—Martindale test. * denotes the Brown–Forsythe correction of F-statistic, n.s. not significant data.

Textile	Test	ANOVA Results			Fisher LSD Test	
		F	P	ω²	F	p
F1	T1	27.069 *	<0.000001	0.782	76.610	<0.000001
	T2	2.655 *	0.0698	0.222	n.s.	n.s.
	MT	6.926 *	0.0052	0.518	5.387	0.0179
F2	T1	8.475 *	<0.0001	0.415	26.452	<0.00001
	T2	4.012	0.0024	0.244	7.869	0.0036
	MT	1.084	0.4241	0.026	n.s.	n.s.
F3	T1	3.953	0.0074	0.467	0.529	0.2382
	T2	5.546 *	0.0081	0.484	2.586	0.0614
	MT	0.989 *	0.5298	0.018	n.s.	n.s.

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
