# Peer review of "Changes in Fabric Surface Pilling under Laser Ablation"

_sensors, 2020, doi:10.3390/s20205832_

Round 1
Reviewer 1 Report
Abstract is too long, please revise.
Keywords shoul be revised to “laser, textile modification, pilling, fractal dimension, optical coherent tomography, computer image analysis”
Barring a few grammatical mistakes, the language is good.
Conclusion section is just a repetition of abstract, please rewrite.
Authors should discuss, in introduction section, briefly smart textiles and cite some related articles;
V. Thekkekara et al. Scientific reports 9, no. 1 (2019): 1-7.
S. Abdelrahman et al. ChemistrySelect 4, no. 13 (2019): 3811-3816.
Micus et al. Materials 13, no. 11 (2020): 2429.
Wang et al. ACS nano 14, no. 3 (2020): 3219-3226.
Ahmed et al. Cellulose 27, no. 5 (2020): 2913-2926.
Mondal et al. Applied thermal engineering 28, no. 11-12 (2008): 1536-1550.
Cherenack et al. Advanced materials 22, no. 45 (2010): 5178-5182.
S. Abdelrahman et al. Journal of Molecular Structure (2020): 128301.
Stoppa et al. sensors 14, no. 7 (2014): 11957-11992.
Author Response
Dear Reviewer,
We would like to thank you very much for your time you spent on correcting our paper and suggestions. They have helped us to improve the paper.
Below we have written answers for your comments. All of them are placed directly under the comments and they are highlighted in the text:
1) Abstract is too long, please revise.
Abstract has been shortened
2) Keywords should be revised to “laser, textile modification, pilling, fractal dimension, optical coherent tomography, computer image analysis”
Keywords have been changed
3) Barring a few grammatical mistakes, the language is good.
English has been checked by the native speaker with technical education
4) Conclusion section is just a repetition of abstract, please rewrite.
Abstract has been changed
5) Authors should discuss, in introduction section, briefly smart textiles and cite some related articles;
- V. Thekkekara et al. Scientific reports 9, no. 1 (2019): 1-7.
- S. Abdelrahman et al. ChemistrySelect 4, no. 13 (2019): 3811-3816.
- Micus et al. Materials 13, no. 11 (2020): 2429.
- Wang et al. ACS nano 14, no. 3 (2020): 3219-3226.
- Ahmed et al. Cellulose 27, no. 5 (2020): 2913-2926.
- Mondal et al. Applied thermal engineering 28, no. 11-12 (2008): 1536-1550.
- Cherenack et al. Advanced materials 22, no. 45 (2010): 5178-5182.
- S. Abdelrahman et al. Journal of Molecular Structure (2020): 128301.
- Stoppa et al. sensors 14, no. 7 (2014): 11957-11992.
The following references has been placed in the text:
- Wang H., Wang H., Wang Y., Su X., Wang C., Zhang M., Jian M., Xia K., Liang X., Lu H., Li S., Zhang Y., Laser Writing of Janus Graphene/Kevlar Textile for Intelligent Protective Clothing, ACS Nano 2020 14 (3), 3219-3226 DOI: 10.1021/acsnano.9b08638
- Micus, S.; Haupt, M.; Gresser, G.T. Soldering Electronics to Smart Textiles by Pulsed Nd:YAG Laser. Materials 2020, 13, 2429.
Additionally, the following text has been added to the Introduction:
The phenomenon of pilling is important not only for aesthetic reasons, but also due to changes in the surface morphology, which is very important in the case of modern technologies related to wearable electronics. The production of new types of materials and the modification of textiles is currently an explored area, especially in the context of using them as a substrate for elements of wearable electronics or textronics [9] or the production of wearable sensors [10,11].”
- Thekkekara L. V., Gu M. Large-scale waterproof and stretchable textile-integrated laser- printed graphene energy storages, Scientific reports 9, no. 1 (2019): 1-7. Doi: https://doi.org/10.1038/s41598-019-48320-z
- Abdelrahman M. S., Fouda M. G., Ajarem J. S., Maodaa S. M., Allam A. A., Khattaba T.A., Development of colorimetric cotton swab using molecular switching hydrazone probe in calcium alginate, Journal of Molecular Structure, 1216, 2020, 128301, https://doi.org/10.1016/j.molstruc.2020.128301
- Lada-Tondyra E., Jakubas, A. Modern Applications of Textronic Systems, Przeglad Elektrotechniczny, 94, 2018, 12 198-201
and suggested references have been cited in the manuscript.
Thank you very much for your time reading our manuscript and all comments.
All numbers of references have been corrected.
Attached please find the improved and upgraded text of our article.
Reviewer 2 Report
Board comments,
The manuscript is a exhaustive study about the influence of the laser power on the laser treated textiles product pilling. Laser treated textiles were correctly analysed via innovated methods as, Martindale’s test and optical coherence tomography. Besides, the pilling of the textiles were adequate analysed through the excellent evaluation of the data with appropriate algorithms.
Although this document reaches its main objective, it is recommended to consider the next comments regarding the paper:
Used laser device details (e.g. power, pulse length and scanning speed) and the types of the textiles products should be commented in the 1. Introduction part. This could increase the interest of the readers to manuscript.
The details of the method and device (supplier and model) used to measure the density of the textiles products should be indicated in 2.1. Textile characteristic part. This could be of the interest of readers.
The supplier of the textiles products should be detailed to increase the impact of the manuscript.
The model, supplier, software of the laser devices and laser processing atmosphere should be indicated 2.2. Process of Laser Modification part. This can be of interest to readers.
The facility details (e.g. speed sliding, supplier and model) used to carry out the T1 and T2 tests should be described in 2.3 Methods of pilling assessment part. This might be of interest to readers.
The supplier of the LSD Fisher should be indicated in the 2.5. Statistical analysis part. This could be of interest to readers.
The conclusion of 389-390 line "The action of the laser beam on the fabric caused local melting on the surface of the fibers of the threads from which the textiles are made." should be supported by any comment or paragraph from 2. Results part.
Specific comments.
My specific commentaries are bellow:
To add reference in 44 line "even to prevent theft [REF]. The possibility of combining several processes, such as surface modification,"
To include reference in 46 line "technologies in the clothing industry[REF]."
To incorporate reference in 55 line "in the fabric structure [REF]."
To add reference in 58 line "determined by touch [REF]. The feel of a fabric has a direct impact on the attractiveness of the product."
To include reference in 70 line "greater [REF]. The denser the structure of the product, the lower the susceptibility to pilling."
To incorporate reference in 1 equation (151 line)
To add reference in 196 line "Martindale’s test MT, carried out in accordance with the PN-EN ISO 12945-2: 2002 standard [REF]"
To include reference in 204 line "Figure 4. Pilling scale as described in PN-EN ISO 12945-2:2002 [REF]"
To incorporate references in 282 line "homogeneity of variance, using the Shapiro-Wilk test [REF] and the Brown-Forsythe test [REF], consecutively, at"
To add reference in 370 line "their low nanocrystallinity [REF]. In the manual tests, a reduction of the pixel fraction of outlying fibers was"
To include reference in 380 line "modified by laser ablation [REF]. However, this phenomenon decreased with increasing laser power, as"
Author Response
Dear Reviewer,
We would like to thank you very much for your time you spent on correcting our paper and suggestions. They have helped us to improve the paper.
Below we have written answers for your comments. All of them are placed directly under the comments and they are highlighted in the text:
The manuscript is a exhaustive study about the influence of the laser power on the laser treated textiles product pilling. Laser treated textiles were correctly analysed via innovated methods as, Martindale’s test and optical coherence tomography. Besides, the pilling of the textiles were adequate analysed through the excellent evaluation of the data with appropriate algorithms.
Although this document reaches its main objective, it is recommended to consider the next comments regarding the paper:
- Used laser device details (e.g. power, pulse length and scanning speed) and the types of the textiles products should be commented in the 1. Introduction part. This could increase the interest of the readers to manuscript.
The information has been placed in the text:
“We used SPI Nd-YAG laser with power up to 18 W, pulse duration 220ns and speed of scanning up to 2000 mm/s.”
- The details of the method and device (supplier and model) used to measure the density of the textiles products should be indicated in 2.1. Textile characteristic part. This could be of the interest of readers.
All possible textile parameters are listed in the Table 1. There should be surface mass of the textile products and it was given by the manufacturer.
- The supplier of the textiles products should be detailed to increase the impact of the manuscript.
All possible, known textile parameters are listed in the Table 1. The described behavior is typical for the presented type of fabrics, regardless of the manufacturer, so in authors’ opinion it is not necessary to detail the supplier of the tested fabric.
- The model, supplier, software of the laser devices and laser processing atmosphere should be indicated 2.2. Process of Laser Modification part. This can be of interest to readers.
The proper information is placed in the text
- The facility details (e.g. speed sliding, supplier and model) used to carry out the T1 and T2 tests should be described in 2.3 Methods of pilling assessment part. This might be of interest to readers.
The speed sliding applied in both tests was about 23-25 cm/s. No other parameters can be determined for our manual tests.
- The supplier of the LSD Fisher should be indicated in the 2.5. Statistical analysis part. This could be of interest to readers.
Reference to the method description has been added
- The conclusion of 389-390 line "The action of the laser beam on the fabric caused local melting on the surface of the fibers of the threads from which the textiles are made." should be supported by any comment or paragraph from 2. Results part.
The mentioned behavior is typical in the case when the laser beam power is too high, and it is the beginning of cutting the fabric. The energy is dissipated on the surface what can been seen as the melted area of the fabric.
Specific comments.
My specific commentaries are below:
- To add reference in 44 line "even to prevent theft [REF]. The possibility of combining several processes, such as surface modification,"
The proper reference has been placed in the text
- To include reference in 46 line "technologies in the clothing industry [REF]."
The proper reference has been placed in the text
- To incorporate reference in 55 line "in the fabric structure [REF]."
The proper reference has been placed in the text
- To add reference in 58 line "determined by touch [REF]. The feel of a fabric has a direct impact on the attractiveness of the product."
The proper reference has been placed in the text
- To include reference in 70 line "greater [REF]. The denser the structure of the product, the lower the susceptibility to pilling."
The proper reference has been placed in the text
- To incorporate reference in 1 equation (151 line)
The equation refers to Figure 2, so the reference has been added in the title of Figure 2
- To add reference in 196 line "Martindale’s test MT, carried out in accordance with the PN-EN ISO 12945-2: 2002 standard [REF]"
The proper reference has been placed in the text
- To include reference in 204 line "Figure 4. Pilling scale as described in PN-EN ISO 12945-2:2002 [REF]"
The proper reference has been placed in the text
- To incorporate references in 282 line "homogeneity of variance, using the Shapiro-Wilk test [REF] and the Brown-Forsythe test [REF], consecutively, at"
The proper reference has been placed in the text
- To add reference in 370 line "their low nanocrystallinity [REF]. In the manual tests, a reduction of the pixel fraction of outlying fibers was"
This part of the sentence was deleted.
- To include reference in 380 line "modified by laser ablation [REF]. However, this phenomenon decreased with increasing laser power, as"
This is authors’ conclusion.
Thank you very much for time reading our manuscript and all comments.
All numbers of references have been corrected.
Attached please find the improved and upgraded text of our article.
Reviewer 3 Report
The paper is interesting and presents valuable experimental findings. It is publishable after the minor revision.
Remarks
- Figures 3, 8 need distinct scale bars supplied for all of the images.
- I will be happy to see the clear definition of the fractal dimension in the revised version of the manuscript. Obviously, we have no a true fractal in the fabrics, reported in the manuscript. The authors deal with the pseudo-fractal. So, the clear definition of the fractal dimension will be helpful for a reader.
- I also recommend to discusss in the revised version of the manuscript the change in the wettability of the textile induced by the plasma ablation. This point is very important from the applicative point of view, when we speak about the textile fabrics; see:Towards understanding hydrophobic recovery of plasma treated polymers: Storing in high polarity liquids suppresses hydrophobic recovery, Applied Surface Science, 273 (2013) 549-553. 4. In the text: "The greatest reduction in the values for first descriptor, by approx. 50%, and the largest reduction in the second coefficient, by approx. 10%, were obtained on trend lines for one of the tested fabrics treated with laser ablation powers of between 0 W and the maximum value of 14 W, after strong manual abrasion. In the case of textiles based on polyester threads, laser modification of the product surface led to a significant reduction in their tendency for pilling". I do not think that the laser ablation power itself is important for ablation; actually, the specific power related to the area of the treated sample is important.
Author Response
Dear Reviewer,
We would like to thank you very much for your time you spent on correcting our paper and suggestions. They have helped us to improve the paper.
Below we have written answers for your comments. All of them are placed directly under the comments and they are highlighted in the text:
The paper is interesting and presents valuable experimental findings. It is publishable after the minor revision.
Remarks
- Figures 3, 8 need distinct scale bars supplied for all of the images.
The proper distinct scale bars have been placed in the images
- I will be happy to see the clear definition of the fractal  dimension in the revised version of the manuscript. Obviously, we have no a true fractal in the fabrics, reported in the manuscript. The authors deal with the pseudo-fractal. So, the clear definition of the fractal dimension will be helpful for a reader. 
The fractal dimension used in the paper is the box-counting (Minkowski) dimension . The proper information and reference has been added to the paper
- Wu J.; Jin X.; Mi S.; Tang J., An effective method to compute the box-counting dimension based on the mathematical definition and intervals, Results in Engineering, 2020, 6, 100106 doi: 10.1016/j.rineng.2020.100106
The description has been added to the paper.
- I also recommend to discuss in the revised version of the manuscript the change in the wettability of the textile induced by the plasma ablation. This point is very important from the applicative point of view, when we speak about the textile fabrics; see:
Towards understanding hydrophobic recovery of plasma treated polymers: Storing in high polarity liquids suppresses hydrophobic recovery,   Applied Surface Science, 273 (2013) 549-553.
To mention plasma ablation the following sentence has been inserted into the text. Laser as well as plasma ablation can be also used to change the wettability of the textile [6].
The mentioned reference has been placed in the text:
- Bormashenko E., Chaniel G., Grynyov R., Towards understanding hydrophobic recovery of plasma treated polymers: Storing in high polarity liquids suppresses hydrophobic recovery Applied Surface Science, 273 (2013) 549-553.
- In the text: "The greatest reduction in the values for first descriptor, by approx. 50%, and the largest reduction in the second coefficient, by approx. 10%, were obtained on trend lines for one of the tested fabrics treated with laser ablation powers of between 0 W and the maximum value of 14 W, after strong manual abrasion. In the case of textiles based on polyester threads, laser modification of the product surface led to a significant reduction in their tendency for pilling". I do not think that the laser ablation power itself is important for ablation; actually, the specific power related to the area of the treated sample is important.   
The abstract has been rewritten so that sentence has been removed.
Thank you very much for time reading our manuscript and all comments.
All numbers of references have been corrected.
Attached please find the improved and upgraded text of our article.